# Computational Modeling of the Interaction of Molecular Oxygen with the miniSOG Protein—A Light Induced Source of Singlet Oxygen

Igor Polyakov [1,2], Anna Kulakova [1] and Alexander Nemukhin [1,2,*]

1   Department of Chemistry, M.V. Lomonosov Moscow State University, Moscow 119991, Russia; polyakoviv@gmail.com (I.P.); kulakovaam@gmail.com (A.K.)
2   N.M. Emanuel Institute of Biochemical Physics, Russian Academy of Sciences, Moscow 119334, Russia
*   Correspondence: anem@lcc.chem.msu.ru

**Abstract:** Interaction of molecular oxygen $^3O_2$ with the flavin-dependent protein miniSOG after light illumination results in creation of singlet oxygen $^1O_2$ and superoxide $O_2^{\bullet-}$. Despite the recently resolved crystal structures of miniSOG variants, oxygen-binding sites near the flavin chromophore are poorly characterized. We report the results of computational studies of the protein−oxygen systems using molecular dynamics (MD) simulations with force-field interaction potentials and quantum mechanics/molecular mechanics (QM/MM) potentials for the original miniSOG and the mutated protein. We found several oxygen-binding pockets and pointed out possible tunnels bridging the bulk solvent and the isoalloxazine ring of the chromophore. These findings provide an essential step toward understanding photophysical properties of miniSOG—an important singlet oxygen photosensitizer.

**Keywords:** miniSOG protein; molecular oxygen; singlet oxygen; molecular dynamics; QM/MM

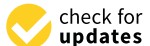



## 1. Introduction

Experimental and theoretical studies of reactions of proteins with molecular oxygen constitute one of the most demanding subjects in life sciences. The chemically inert molecule $O_2$ in the ground triplet spin state ($^3O_2$) can be activated inside protein cavities to initiate reactions, which often result in creation of reactive oxygen species, such as singlet oxygen $^1O_2$, superoxide $O_2^{\bullet-}$, and hydrogen peroxide $H_2O_2$ [1,2].

In this work, we considered a small flavin-dependent protein miniSOG (singlet oxygen generator), which was created as a light-induced source of singlet oxygen [3]. The miniSOG protein was employed in the correlated light and electron microscopy as a photosensitizer for cancer cells and as a marker for fluorescence imaging. Understanding photophysical properties of miniSOG requires the knowledge of structures of the protein, preferably with trapped oxygen molecules. Reference [4] reports the first crystal structure of miniSOG deposited to the Protein Data Bank [5], PDB ID 6GPU. It contains the flavin mononucleotide (FMN) cofactor and the chloride ion, which is assumed to mimic molecular oxygen in the cavity of the originally engineered protein. Reference [6] reports crystal structures of other miniSOG variants, namely structures of mutants R57Q, Q103L, and structures, in which FMN is replaced by a shorter cofactor riboflavin (RF). Figure 1 shows the chemical formulae of FMN and RF, as well as the common numbering of atoms in the chromophore—the isoalloxazine ring of FMN or RF.

The goal of the introduced changes [6] in the structure of the original miniSOG was to increase the quantum yield of singlet oxygen generation $\Phi_\Delta$ observed in the initial variant. One of the reasons for its lower $\Phi_\Delta$ (0.03 vs. 0.51 for FMN in solutions) was attributed to a restricted $^3O_2$ access to the isoalloxazine ring of FMN inside the protein.

Thus, the replacements aimed to expand the oxygen tunnels from the bulk solvent leading to the chromophore.

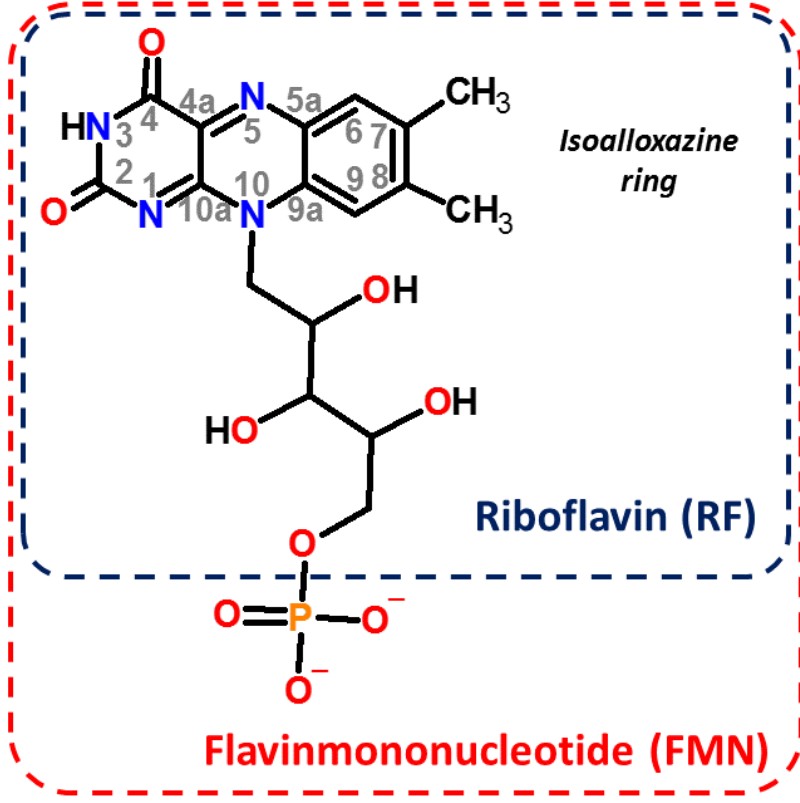

**Figure 1.** The chemical formulae of FMN and RF and the conventional designation of the atoms in the isoalloxazine ring.

We comment that localization of dioxygen in protein cavities is a nontrivial experimental task. Only a few structures in the Protein Data Bank [5] report tentative complexes of molecular oxygen near non-metallic sites in proteins, e.g., [7–9]. The revealed location of the chloride ion in the structures [4] does not favor its close contact with the chromophore to account for efficient protein−oxygen interaction (see Materials and Methods below). The authors claim that locations of the chloride anion in crystal structures "could serve as transient binding sites for $^3O_2$, between which it could hop before reaching the vicinity of the isoalloxazine ring" [6] (p. 1548).

Molecular dynamics (MD) computer simulations can be used to disclose oxygen diffusion pathways in proteins [10–15] and to obtain starting structures for prospective quantum-based calculations of reactions of proteins with molecular oxygen [8,9,15–18]. The present work aims at predicting oxygen-binding sites in the miniSOG variants using MD simulations with force-field interaction potentials and with quantum mechanics/molecular mechanics (QM/MM) potentials.

## 2. Materials and Methods

The crystal structure PDB ID 6GPU [4] was used as a source of coordinates of heavy atoms to construct all-atom molecular model systems of miniSOG with the FMN cofactor, whereas the structure PDB ID 7QF4 [6] was used to build model systems of the miniSOG mutant R57Q/Q103L with the RF cofactor. We denoted the corresponding models as miniSOG[FMN] and miniSOG[RF], respectively. It should be noted that the experimental paper [6] reports the results for several protein variants, including single mutants, R57Q and Q103L, as well as the original protein with FMN replaced by RF. In our work, we only considered an extreme case (called here miniSOG[RF]) combining all replacements. We

intended to compare dynamical properties of the protein−oxygen systems for close but different protein compositions.

Figure 2 illustrates the important features of the models. The secondary structure of the proteins with a cofactor (FMN or RF) revealed in the X-ray studies [4,6] is shown in the central part of Figure 2. Positions of the pairs of amino acid residues, which differ in the miniSOG[FMN] and miniSOG[RF] systems (R57/Q57 and Q103/L103), are identified. The chloride ion resolved in the crystal structure PDB ID 6GPU stays rather far from the chromophore; in particular, the distance from Cl⁻ to the closest atom of the isoalloxazine ring C9 is 4.60 Å. We also show positions of the T28 and T48 amino acid residues, which feature the flexible loop 28–48.

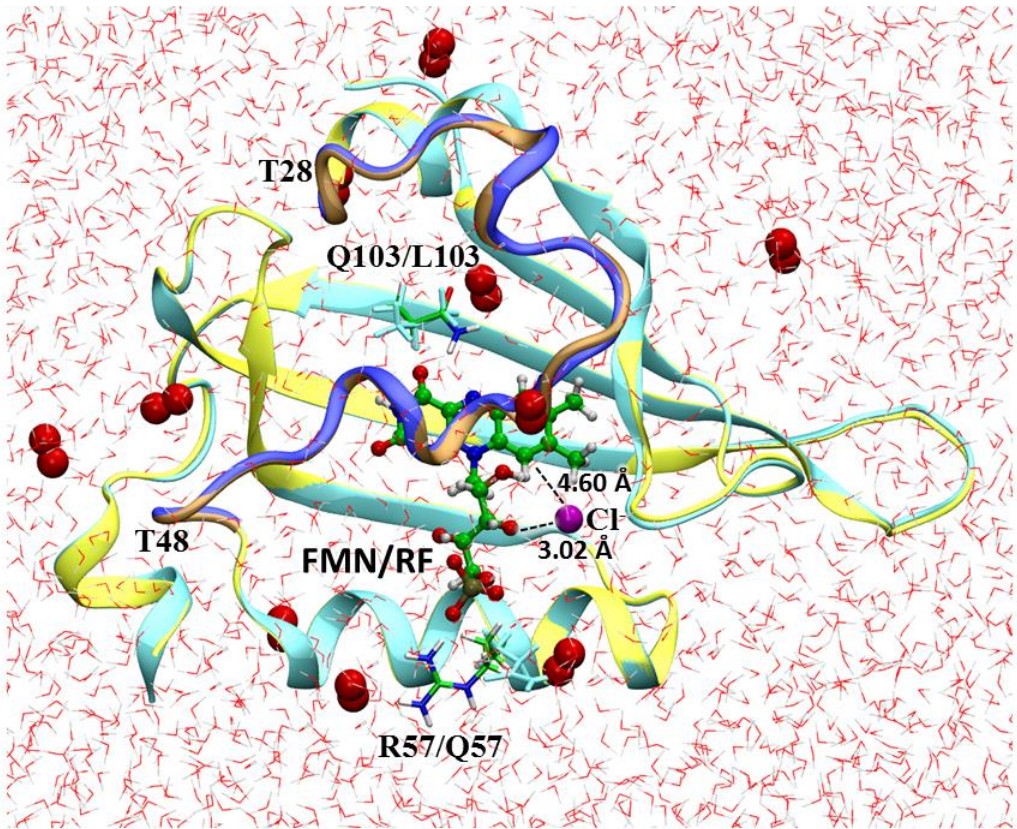

**Figure 2.** A view on the model systems created following the crystal structures. Yellow and orange colors refer to the structure of PDB ID 6GPU; cyan and blue colors refer to the structure of PDB ID 7QF4. In this and the other figures, carbon atoms are shown in green, oxygen atoms are shown in red, nitrogen atoms are shown in blue, phosphorus atoms are shown in orange, chloride ions are shown in magenta, and hydrogen atoms are shown in white. Water-solvent shells surrounding the proteins are shown by red and white lines, whereas the initial positions of the oxygen molecules for subsequent MD simulations are shown as the pairs of large red balls.

When constructing miniSOG[FMN] and miniSOG[RF] model systems, hydrogen atoms were added, assuming the conventional protonation states of the polar residues at neutral pH: Arg and Lys were charged positively; Glu and Asp were charged negatively; the His85 residue was assumed in the neutral state. Water molecules resolved in the crystal structures were kept in the model systems. Solvation water boxes were built using the visual molecular dynamics (VMD) program [19]. Figure 2 shows water solvation shells around the protein macromolecule. The systems were neutralized by adding sodium ions. Ten dioxygen molecules were placed at random places close to the surface of the protein as shown in Figure 2. The CHARMM36 force field topology and parameters [20] were employed along with the TIP3P parameters for water molecules. Parameters of FMN and RF in the oxidized form of flavin were taken from reference [21], and parameters of

molecular oxygen were taken from reference [22]. In total, the miniSOG[FMN] system contained 20,878 atoms, and the miniSOG[RF] system contained 17,509 atoms.

Classical MD trajectories were simulated using the NAMD 3.0 software package [23]. The isothermal-isobaric (NPT) ensemble at P = 1 atm and T = 300 K was employed with the Nosé−Hoover Langevin piston pressure control and the Langevin dynamics. Periodic boundary conditions and the particle-mesh Ewald algorithm to account for the long-range electrostatic interactions were applied, whereas the non-bonded interaction cut-off parameter was set to 12 Å and the integration step was set to 2 fs with the SHAKE/SETTLE algorithms applied to constrain bonds to hydrogen atoms. A harmonic constraint potential of 0.1 kcal/$Å^2$ was applied to the CA atoms of the beta sheets of the protein. A total of 3300 ns from 5 trajectories was produced for the miniSOG[FMN] model, and a total of 7555 ns from 6 trajectories was produced for the miniSOG[RF] model.

As described in Results, several binding pockets in protein cavities were identified in classical MD simulations for the miniSOG[FMN] and miniSOG[RF] systems. Two of these pockets (called below as A and B) were common for both systems. Pocket A was particularly interesting, because the oxygen molecule resided above and close to the isoalloxazine ring. To investigate this binding site at a higher accuracy level, we carried out MD simulations with the quantum mechanics/molecular mechanics (QM/MM) potentials for the miniSOG[RF] system in the vicinity of pocket A. The entire riboflavin molecule and the oxygen molecule were assigned to the QM subsystem, whereas the protein and solvent shells constituted the MM part. The energies and energy gradients in QM were computed using the density functional theory (DFT) level with the range-separated wb97X functional [24] with the D3 dispersion correction [25]. The 6-31G** basis set was employed. The unrestricted DFT approach was used to describe the triplet spin state of the QM subsystem due to the ground electronic state of $^3O_2$. The MM part was described by the CHARMM36 force field [20]. The trajectories were computed using the software stack of NAMD [26] and TeraChem [27]. The trajectory of a total length of about 100 ps (with a 1 fs integration timestep) was produced in these runs. The MDAnalysis software [28] was employed to analyze molecular dynamics trajectories.

The data extracted from the MD trajectories and used for the analysis are available in the CVS format at the general-purpose open-access repository ZENODO (accessed via https://doi.org/10.5281/zenodo.7722156 (accessed on 11 March 2023)).

## 3. Results

As written in Materials and Methods, 10 dioxygen molecules were randomly placed close to the surface of the protein (see Figure 2), and several independent classical MD trajectories were executed starting from the created model systems for both miniSOG[FMN] and miniSOG[RF] proteins. Two out of eleven runs did not lead to the diffusion of oxygen molecules inside the protein; while the other attempts resulted in the penetration of dioxygen inside the protein macromolecules and the formation of the oxygen-binding pockets. Figure 3 summarizes the results of classical MD simulations. The binding pockets A and B, which are highlighted in yellow in Figure 3, were almost similar in both model systems miniSOG[FMN] and miniSOG[RF].

To reach pocket A in miniSOG[FMN], formed by the isoalloxazine ring and the V6, I17, F24, F42, and L43 amino acid residues, the oxygen molecule entered through a cluster of T8, P10, N15, and F99 residues. The oxygen molecule could stay for a long time inside the cavity but occasionally could escape through the gate near the T48 position. Pocket A seemed to be the most populated in the miniSOG[RF] system: the diffusion pathway predominantly included the F24, Y30, I35, F42, and L43 residues.

Pocket B also showed that the oxygen molecule could reside near the isoalloxazine ring inside the hydrophobic cavity; however, at larger distances from the oxygen atoms to the functionally important flavin atoms (C4a, N5), pockets C, D, and E were populated differently in both model systems.

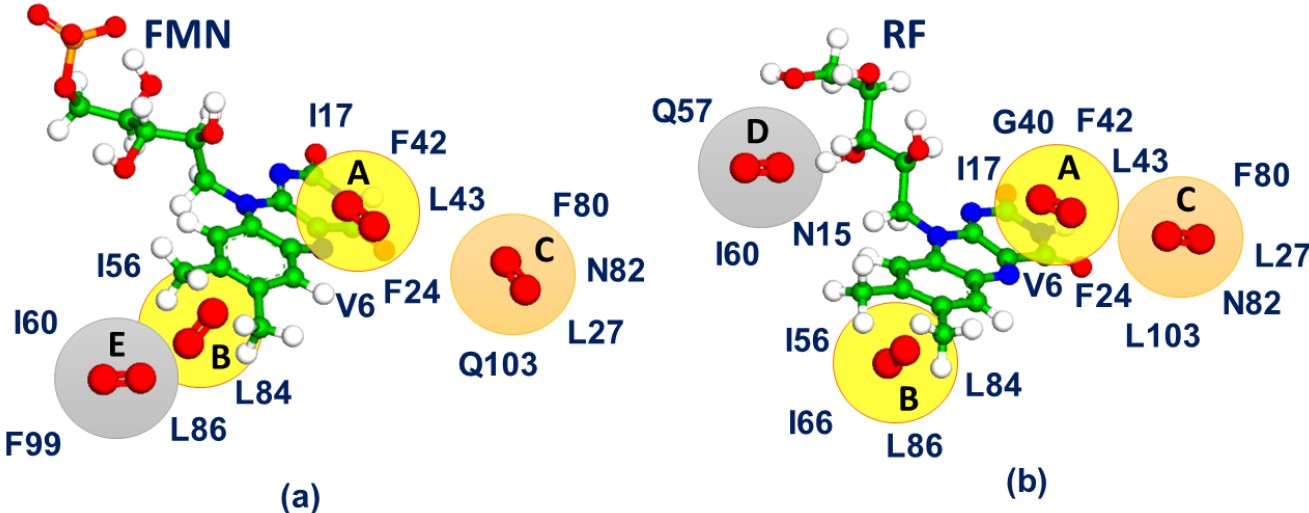

**Figure 3.** The located oxygen-binding pockets in the miniSOG[FMN] (**a**) and miniSOG[RF] (**b**) model systems.

We applied the MDAnalysis toolkit [28] to reveal the most important features observed along computed trajectories, which are summarized in Table 1 and in Figures 4–8. We skipped the N- and C-terminus regions (residues 1–4 and 106–116) in our analysis due to the high dynamics flexibility in these regions. In addition, the first 10 ns period out of each trajectory length was excluded from the analysis. The alignment in the RMSD (root-mean-square deviation) calculations was conducted on the backbone of protein residues 5–105. The calculated RMSDs for the backbone of residues 5–105 (column "BB 5–105" in Table 1), the backbone of the loop residues (column "BB 28–48"), all heavy atoms of the loop residues ("28–48 noH") and the isoalloxazine ring ("ISO") of the FMN and RF moieties are reported in Table 1 for both the considered molecular systems.

**Table 1.** RMSD analysis. Values in Å are shown in the format: median ± standard deviation.

| Model System | BB 5–105 | BB 28–48 | 28–48 noH | ISO |
|---|---|---|---|---|
| miniSOG[FMN] | 0.8 ± 0.2 | 0.7 ± 0.2 | 1.4 ± 0.3 | 0.9 ± 0.2 |
| miniSOG[RF] | 1.1 ± 0.2 | 1.1 ± 0.3 | 2.0 ± 0.3 | 1.0 ± 0.2 |

These results indicate that the R57Q and Q103L mutations as well as the replacement of the flavin moiety led to higher overall backbone flexibility, including the loop region. This is especially evident from the RMSD analysis of all heavy atoms (column "28–48 noh") of the loop residues. On the contrary, the isoalloxazine ring was rather stable for both the original miniSOG[FMN] and mutated miniSOG[RF] model systems (column "ISO"), although it was slightly more flexible in the mutant.

The RMSD analysis failed to show which residues mostly contributed to the increased motion of the loop; therefore, the root-mean-square fluctuation analysis (RMSF) was conducted as illustrated in Figures 4–6.

The data presented in Figures 4–6 provide further insights into the differences of protein dynamics. The miniSOG[RF] system was clearly more flexible than the miniSOG[FMN] system in the region of protein residues 10–15 and in the loop region. In the miniSOG[FMN] system, the first region constituted the loop between the β-strands, where the D14 side chain formed a salt bridge with the R41 side chain, which, in turn, formed a contact with the phosphate group of the FMN. In the miniSOG[RF] system, the phosphate group was absent, weakening the R41−D14 salt bridge. This resulted in a significant fluctuation of the 10–15 loop and of the region of residues 39–42 (Figure 6). Finally, the important 28–48 loop region was affected, which influenced oxygen-binding pocket A.

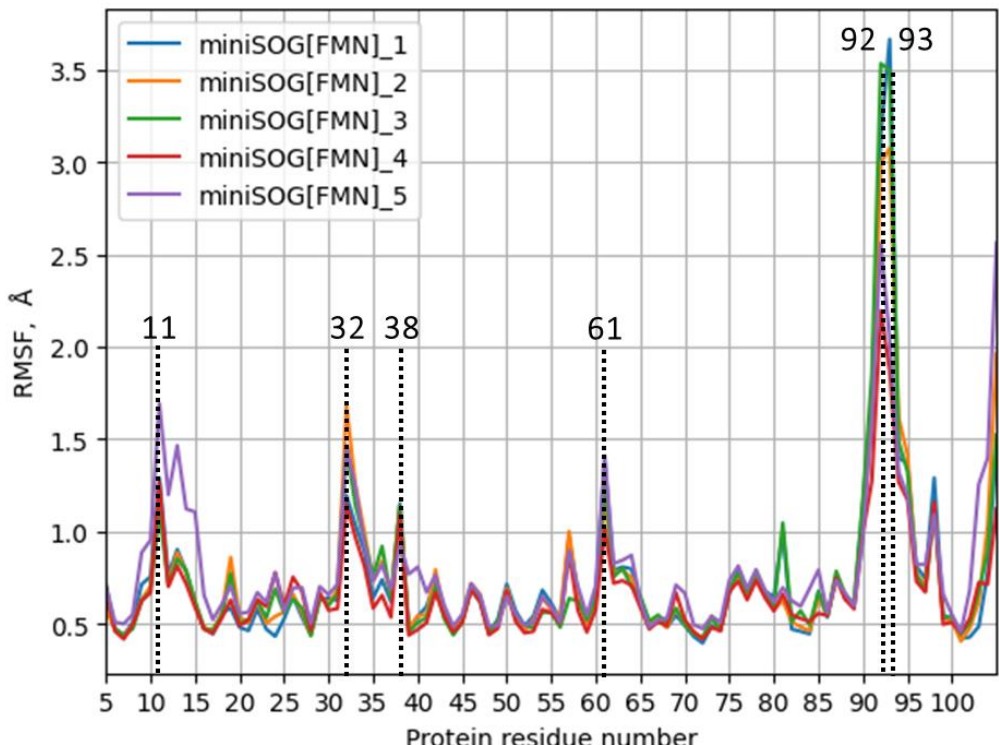

**Figure 4.** RMSF values for the 6 classical MD trajectories for the miniSOG[FMN] system.

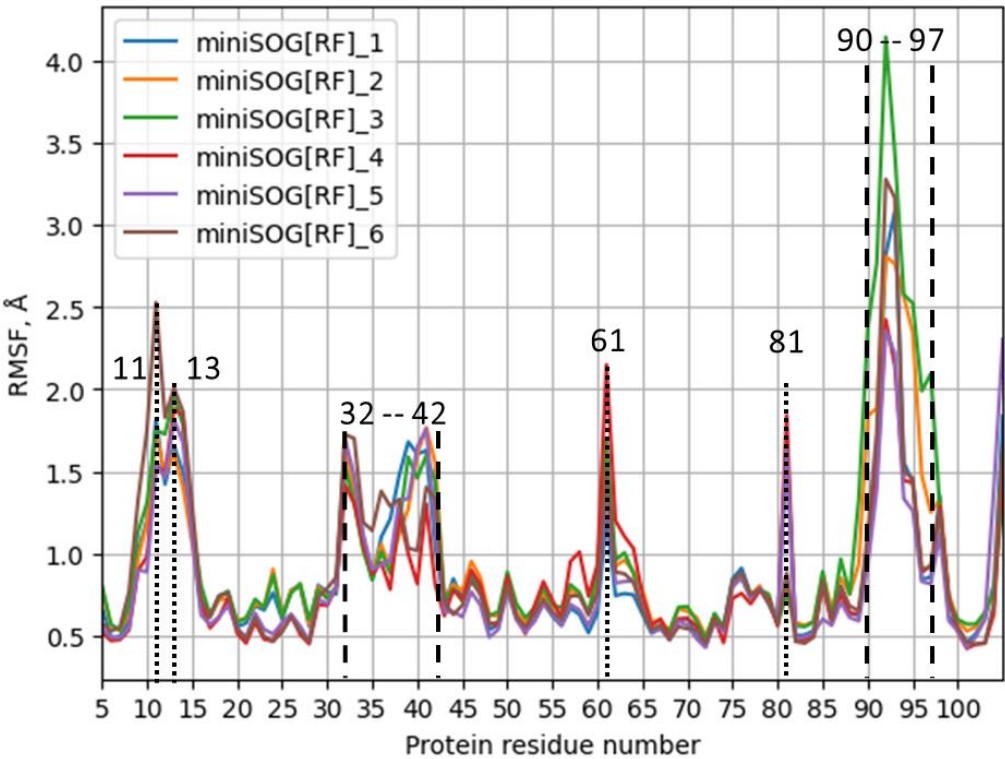

**Figure 5.** RMSF values for the 6 classical MD trajectories for the miniSOG[RF] system.

Figures 4 and 5 show that the RMSF variance between the trajectories in the 28–48 loop region was higher for the miniSOG[RF] system, while for both the considered proteins, the highest variance was in the residues 90–97 region. The latter referred to another loop between the β-strands, which was located near the 10–15 loop. For the miniSOG[FMN]

system, the 90–97 loop was connected to the 10–15 loop through the Q97−R11 contact, while for the miniSOG[RF] system, no important contacts were observed.

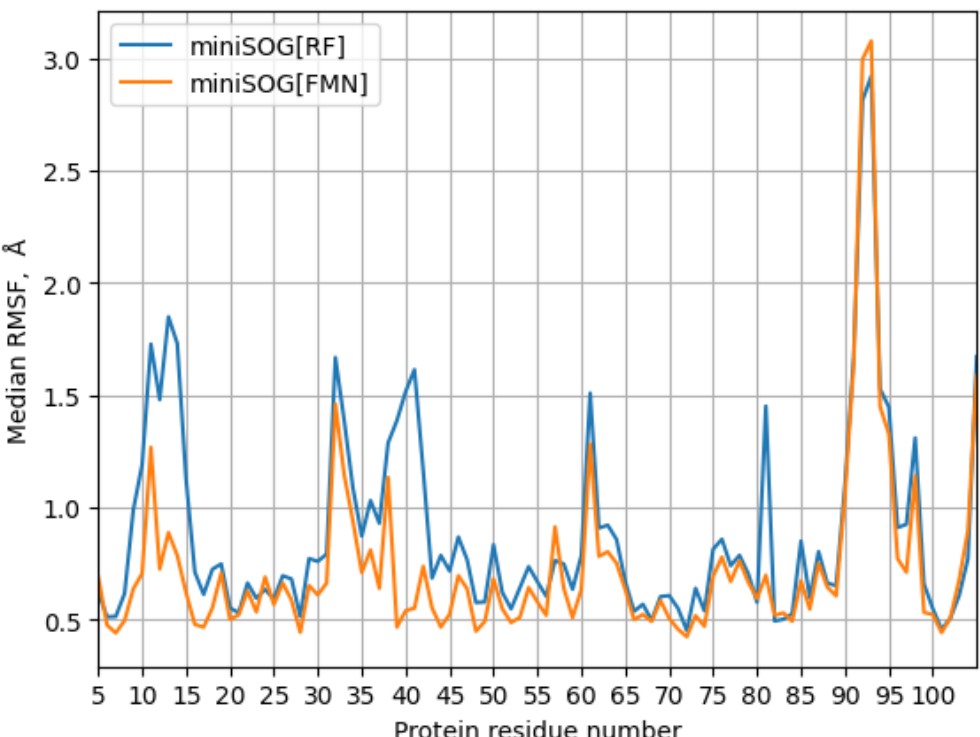

**Figure 6.** RMSF values for the miniSOG[RF] and the miniSOG[FMN] systems averaged over the respective MD trajectories.

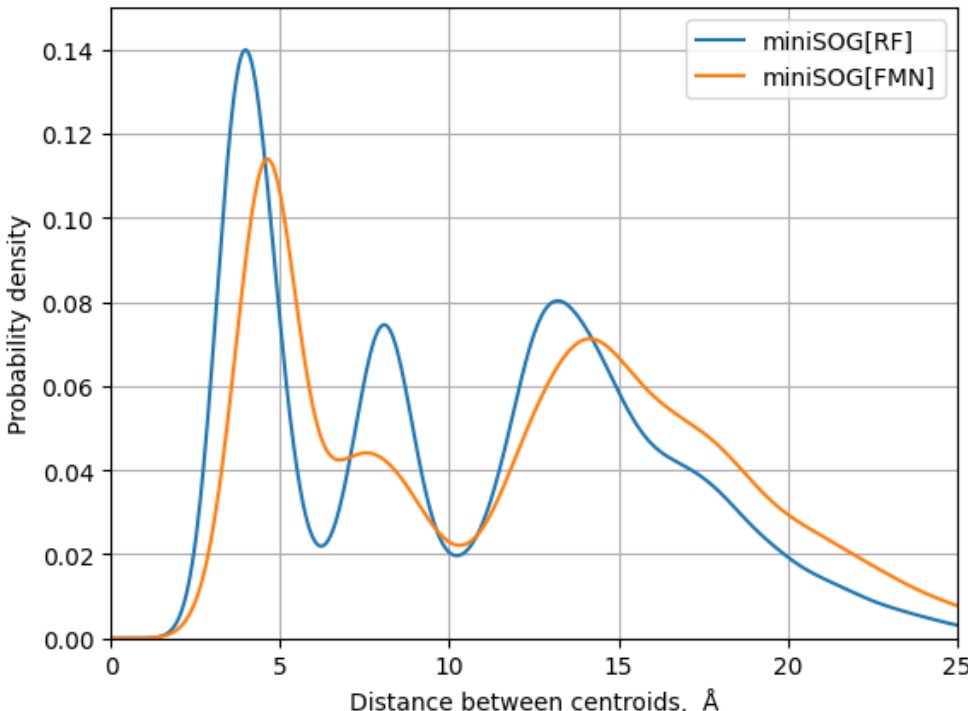

**Figure 7.** Probability density functions for the distribution of the distance between the isoalloxazine ring center and the center of the closest oxygen molecule for 6 combined trajectories for miniSOG[RF] and 5 combined trajectories for miniSOG[FMN].

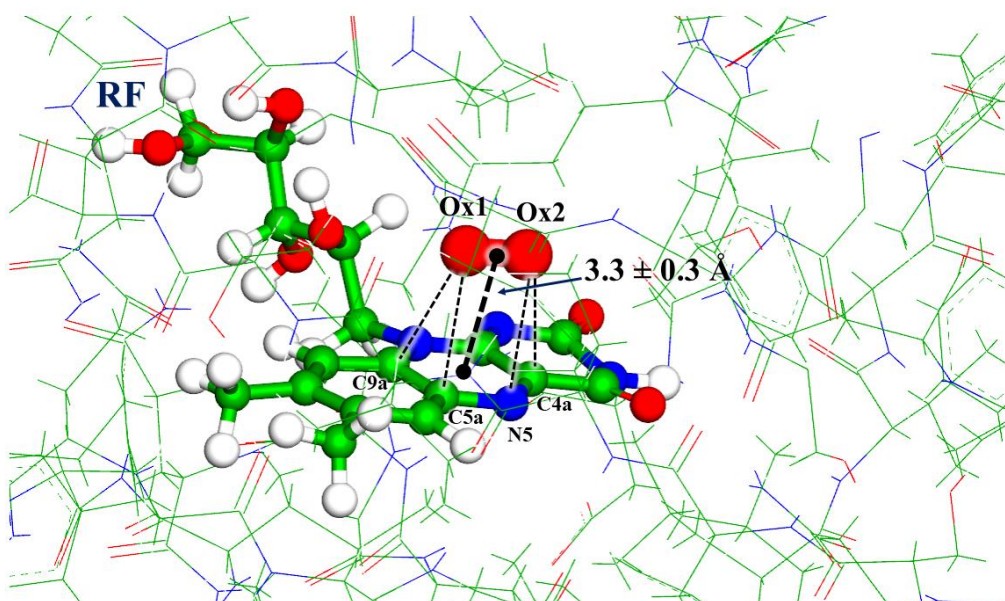

**Figure 8.** The flavin−oxygen complex in the miniSOG[RF] model system observed in a typical frame along the QM/MM MD trajectory in the pocket A area (see Figure 3).

The differences in the trajectories for the original variant miniSOG[FMN] and for the mutated miniSOG[RF] system were best revealed in the analysis of the distribution of oxygen molecules. For a quantitative characterization, we selected a distance $R_c$ between the geometry center of the isoalloxazine ring and the center of the closest oxygen molecule. The distribution of this distance, which can be evaluated for all trajectories, is shown in Figure 7. The first peak of the probability density function around 3–6 Å was mainly associated with oxygen-binding pocket A (see Figure 3) right above the isoalloxazine ring. The peaks in the 6–10 Å region roughly corresponded to other pockets. If we filtered out the values greater 6 Å, we obtained the following distribution: 11% of all trajectory frames for the miniSOG[FMN] system demonstrated a close proximity of oxygen to the isoalloxazine ring with $R_c = 4.6 \pm 0.5$ Å, whereas for the miniSOG[RF] system, 30% of all trajectory frames showed a shorter distance $R_c = 4.0 \pm 0.6$ Å. This result correlates with the experimental observation [6] that the mutated miniSOG is a better singlet oxygen generator.

Due to this reason, we carried out additional MD simulations with the QM/MM potentials for the miniSOG[RF] protein variant. An analysis of an about 100 ps trajectory obtained at the QM/MM MD level showed that the oxygen molecule stably resided in pocket A and it was located closer to the isoalloxazine ring than in the model systems treated by classical MD simulations. As shown in Figure 8, the $R_c$ value in QM/MM MD simulations was $3.3 \pm 0.3$ Å compared to that of $4.0 \pm 0.6$ Å in classical calculations.

## 4. Discussion

The results of the present MD simulations allowed us to identify several oxygen-binding pockets inside the protein macromolecules in the miniSOG[FMN] and miniSOG[RF] model systems. The identified pocket A (see Figures 3 and 8) showed the most important cavity, because the oxygen molecule accommodated right above the isoalloxazine ring, and this position should be considered at the first place in future studies of photo-induced processes in miniSOG. The distances from the oxygen atoms of $O_2$ to the atoms of the ring C4a, N5, which are considered as critical centers for flavin functionalization [2], were within 3.5–4 Å (Figure 8), although we noted that these distances were longer than those in the oxygen−protein complexes observed in proteins containing the reduced form of flavin [2,8,18].

We can also distinguish the original and mutated systems on the accessibility of oxygen to the flavin; the oxygen molecule is significantly more likely to be in a close proximity to isoalloxazine ring in the miniSOG[RF] protein than in the original miniSOG[FMN] construct.

There are plenty of possible pathways for the oxygen molecule to enter the protein and to diffuse to the chromophore. The protein is rather small, stable, and rigid, except for the large loop formed by residues 28–48 between the α-helices that are formed on the edges of the β-strands (see Figure 2). This loop contacts β-strands and forms a bulk of non-polar residues, which form pocket A right above the isoalloxazine ring. Because this loop is dynamically flexible, there are fluctuations in the main and side chains, which influence oxygen diffusion pathways. Thus, amino acid mutations can affect the singlet oxygen generation, as observed in experimental studies [6]. The double mutation considered in the present simulations illustrates this issue: we confirmed that oxygen can enter through the tail-associated pathway (R57Q and the FMN/RF replacements). In addition, the replacement Q103L opens a new pathway for oxygen and indirectly influences dynamics of a very important pocket near the F24−F42 contact. Our calculations and analysis show how mutations and flavin replacement influence the protein dynamics, which results in a better accommodation of an oxygen molecule in pocket A.

From the methodology side, we emphasize that the model systems miniSOG[FMN] and miniSOG[RF] constructed in the present work provide more appropriate structures of the protein with molecular oxygen than the available crystal structures with chloride anion [4]. The latter tends to reside in the solvent accessible protein cavities, whereas molecular oxygen tends to occupy hydrophobic regions.

Another issue is the significance of MD simulations with the QM/MM potentials, which are becoming available recently. The ab initio type approaches in the QM subsystems provide a more accurate description of the protein−oxygen interaction than classical force-field parameters. In the present application, we note that the use of more accurate quantum potentials results in a more close and tight distribution of an oxygen molecule relative to the isoalloxazine ring than using the classical parameters.

## 5. Conclusions

The results of molecular dynamics simulations with force-field and quantum mechanics/molecular mechanics interaction potentials allowed us to localize binding sites of the oxygen molecule near the flavin chromophore for the two variants of the miniSOG protein. The binding site, in which the oxygen molecule is located at the distance about 3.3 Å to the geometry center of the isoalloxazine ring of the chromophore, corresponds to the most promising structure for studies of singlet oxygen generation upon light excitation of miniSOG.

**Author Contributions:** Conceptualization, I.P., A.K. and A.N.; methodology, I.P. and A.K.; validation, I.P., A.K. and A.N; formal analysis, I.P., A.K. and A.N; investigation, I.P., A.K. and A.N; data curation, I.P., A.K. and A.N; writing—original draft preparation, A.N; writing—review and editing, I.P., A.K. and A.N; visualization, I.P. and A.K.; supervision, A.N.; project administration, A.N.; funding acquisition, I.P. and A.K. All authors have read and agreed to the published version of the manuscript.

**Funding:** This research was funded by the Russian Science Foundation (grant number: 22-13-00012).

**Data Availability Statement:** The data extracted from the MD trajectories and used to produce Table 1 and Figures 4–7 are available in the CVS format at the general-purpose open-access repository ZENODO (accessed via https://doi.org/10.5281/zenodo.7722156 (accessed on 11 March 2023)).

**Acknowledgments:** The research was carried out using the equipment of the shared research facilities of high-performance computing resources at Lomonosov Moscow State University. The use of supercomputer resources of the Joint Supercomputer Center of the Russian Academy of Sciences is acknowledged.

**Conflicts of Interest:** The authors declare no conflict of interest.

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
