# Peer review of "Computational Modeling of the Interaction of Molecular Oxygen with the miniSOG Protein—A Light Induced Source of Singlet Oxygen"

_biophysica, doi:10.3390/biophysica3020016_

Round 1

Reviewer 1 Report

Remarks to the authors

In this manuscript, Polyakov et al. examined the interaction between oxygen and miniSOG. The authors used molecular dynamics simulations to characterize the structural flexibility of two miniSOG variants with different chromophores and predict the potential oxygen binding sites. The simulation results show that the FMN/RF replacement and the two mutations – R57Q and Q103L – lead to a higher accessibility of oxygen to the chromophore. Overall, this manuscript is a solid piece of work that provides mechanistic insights into the photophysical properties of. miniSOG.

Below, I have a few comments/questions for the authors:

(1)   In the current manuscript, the authors considered the original miniSOG[FMN] and the mutated miniSOG-R57Q/Q103L[RF]. However, there’re multiple intermediates between these two systems. For example, one can keep the original protein and only replace FMN with RF. Similarly, one can also consider a mutated yet FMN-binding protein (see for example: https://www.rcsb.org/structure/7QF5). Is there any reason why the authors ended up looking at the two structures studied?

Note that in the paper by Lafaye et al. (Ref.[6] of the current manuscript), they reported that individual changes to the original miniSOG[FMN] (either FMN/RF replacement or Q103L mutation) could lead substantial increase in the quantum yield. I’m not requesting this, but I think looking at those intermediate miniSOG variants might clarify the effect of individual changes.

(2)   In the paper by Lafaye et al., they also commented “double mutant miniSOG-R57Q/Q103L was gradually losing its chromophore”. I doubt the MD simulations are long enough to observe these events, but I wonder whether the authors observe, for example, the chromophore is more dynamic in the mutated protein than in the original miniSOG.

(3)   There are several typos/wording issues that the authors need to address. Some examples are:

a.      Lines 32–33 (“Understanding photophysical properties of miniSOG deserves the knowledge of structures of the protein,…”): I think the authors mean “requires” rather than “deserves”?

b.      Lines 236–238 (“The distances from the oxygen atoms of O2 to the atoms of the ring C4a, N5, …, are within 3.5 ÷ 4 Å, …”): Do the authors mean “3.5 – 4 Å”?

c.       Lines 250–252 (“…: we confirm that oxygen can enter through the tail-associated pathway (Q57L and the FMN/RF replacements).”): “Q57L” should be “R57Q”?

Author Response

Please see tha attachment

Reviewer 2 Report

This is an excellent model study concerning the interaction of molecular oxygen with a small flavin-dependant protein. The results strongly support the possibility of singlet oxygen generation by this protein with the Rc between the chromophore and the dioxygen molecule being 3.3 ± 0.3 Å by the author's QM/MM MD simulations. These results should be of interest to researchers in the filed of photobiology and photochemistry. The paper is very well written and concise. I have only very few suggestions for improvement; my only significant recommendation would be for the authors to slightly expand the introduction. They mention that both superoxide anion and singlet oxygen may be produced by this protein. The authors comment on the low singlet oxygen quantum yield, but do not provide any numbers. Is the actual quantum yield known? If yes, perhaps it could be given together with that of the free molecule in solution, in the paragraph on page 2, lines 46-51, together with the appropriate references.  Also, is anything known about the amount of superoxide produced?

Minor typo, page 5, line 173, there should be no comma after "failed to show".

Overall, this is a very good paper and I strongly recommend publication.
